# High-Frequency Dielectrophoresis Reveals That Distinct Bio-Electric Signatures of Colorectal Cancer Cells Depend on Ploidy and Nuclear Volume

**DOI:** 10.3390/mi14091723

**Published:** 2023-09-01

**Authors:** Josie L. Duncan, Mathew Bloomfield, Nathan Swami, Daniela Cimini, Rafael V. Davalos

**Affiliations:** 1Department of Mechanical Engineering, Virginia Tech, Blacksburg, VA 24061, USA; josied@vt.edu; 2Department of Biological Sciences and Fralin Life Sciences Institute, Virginia Tech, Blacksburg, VA 24061, USA; 3Department of Electrical and Computer Engineering, University of Virginia, Charlottesville, VA 22908, USA; 4Department of Biomedical Engineering and Mechanics, Virginia Tech-Wake Forest School of Biomedical Engineering and Sciences, Blacksburg, VA 24061, USA

**Keywords:** electrokinetics, aneuploidy, microfluidics

## Abstract

Aneuploidy, or an incorrect chromosome number, is ubiquitous among cancers. Whole-genome duplication, resulting in tetraploidy, often occurs during the evolution of aneuploid tumors. Cancers that evolve through a tetraploid intermediate tend to be highly aneuploid and are associated with poor patient prognosis. The identification and enrichment of tetraploid cells from mixed populations is necessary to understand the role these cells play in cancer progression. Dielectrophoresis (DEP), a label-free electrokinetic technique, can distinguish cells based on their intracellular properties when stimulated above 10 MHz, but DEP has not been shown to distinguish tetraploid and/or aneuploid cancer cells from mixed tumor cell populations. Here, we used high-frequency DEP to distinguish cell subpopulations that differ in ploidy and nuclear size under flow conditions. We used impedance analysis to quantify the level of voltage decay at high frequencies and its impact on the DEP force acting on the cell. High-frequency DEP distinguished diploid cells from tetraploid clones due to their size and intracellular composition at frequencies above 40 MHz. Our findings demonstrate that high-frequency DEP can be a useful tool for identifying and distinguishing subpopulations with nuclear differences to determine their roles in disease progression.

## 1. Introduction

Changes in DNA content are a fundamental aspect of cancer biology. Aneuploidy, or incorrect chromosome numbers, is found in 75% of hematopoietic and 90% of solid tumors [1,2]. How tumors accumulate high levels of aneuploidy remains enigmatic, since chromosome gains and losses are generally detrimental to cell fitness [3,4]. One possibly favorable path to aneuploidy is via whole-genome doubling (WGD) [5], which produces tetraploid (4N) cells with twice the DNA content of diploid (2N) cells and is among the most-common genomic alterations in cancer [6]. Several studies have shown that WGD+ tumors accumulate more aneuploidy than WGD- tumors [6,7,8,9], and both WGD and aneuploidy are associated with poor patient prognosis and resistance to cancer therapies [3,6,10,11]. Therefore, it is critical to understand how 4N and aneuploid cells develop and evolve in cancers.

The isolation of 4N cells from tumor-derived cancer cell populations would enable studies aimed at understanding the contribution of 4N cancer cells to cancer progression and their sensitivity to chemotherapeutic drugs. Current methods used to identify 4N cells in a population, such as fluorescence-activated cell sorting (FACS) or karyotyping, typically require cell labeling and/or fixation. These procedures result in low sample viability and limit downstream applications. One potential alternative is dielectrophoresis (DEP), a label-free technique that uses electric fields to manipulate and sort particles according to their unique dielectric signature. Since DEP does not require cell fixation or labeling, the cells remain viable and are suitable for analysis after sorting [12]. DEP has been used to manipulate, characterize, and distinguish a wide range of biological particles, such as mammalian cells [12,13,14,15], bacteria [16,17,18,19,20,21], and viruses [22,23,24]. Regarding the separation of cancer cells, Gupta et al. developed the commercially available Apostream^TM^ to fulfill a need in the marketplace for separating circulating tumor cells from blood without labels [25]. Several studies have proposed proof-of-concept approaches to DEP-based separations of dissimilar cell types [26], while others have focused on distinguishing subpopulations from the same cell type [27,28].

DEP can separate cell subpopulations based on their size and/or dielectric dispersion, which describes the change in the electrical permittivity of a cell with frequency. The DEP frequency regime determines what cellular properties are reflected in the dielectric dispersion. In the low-frequency α-regime (<1 kHz), the dielectric dispersion is based on cell membrane properties. In higher-frequency regimes, known as the β-regime (1 kHz–100 MHz), the dielectric dispersion is affected by the internal structures of the cells, such as the cytoplasm and the nucleus and its contents. Beyond the β-regime, the γ-regime (100 MHz–100 GHz) is associated with water molecule polarization.

Compared to low-frequency DEP applications that capitalize on the properties of the plasma membrane, few studies have explored the distinguishing capabilities of the high-frequency regime using DEP. High-frequency DEP requires advanced electronics and is known to suffer capacitive losses, making it difficult to explore novel applications relevant to the cell interior. The authors have previously shown that single-cell impedance cytometry can be used to show the dependence of the impedance phase at 50 MHz on cell subpopulations with double the DNA content in their interphase stage before mitosis [29]. Manczak et al. and Lambert et al. used ultrahigh-frequency electrorotation to characterize the undifferentiated and stemness phenotypes of glioblastoma cells [27,30]. Hjeij et al. showed that HCT116 and SW620, two colorectal cancer (CRC) cell lines of varying aggressiveness and ploidy, have different high-frequency DEP signatures using electrorotation [31]. Although this study suggests high-frequency DEP may be able to detect and sort tumor cell populations with altered DNA content, it is unclear to what extent other underlying biophysical differences, such as cell organelle changes, may also have influenced the high-frequency DEP signatures of these cancer cell lines.

To investigate the functional effects of WGD in cancer cells, we recently generated several isogenic 4N clones from diploid DLD-1 CRC cells and found that cell and nuclear size do not always scale with DNA content following WGD [32]. Because these clones differ in size while maintaining near-4N genomes, these isogenic 2N and 4N CRC cell lines are useful for systematically interrogating the effects of DNA content and nuclear size on high-frequency dielectrophoretic signatures. We investigated 2N cells, two 4N clones similar in size to 2N cells (small clones, denoted as S1 and S2), and two 4N clones larger than 2N cells (large clones, denoted as L1 and L2). We hypothesized that 2N cells and 4N clones will display distinguishable dielectrophoretic behavior at high frequencies due to ploidy and cell and nuclear size. We show that high-frequency dielectrophoresis has applications toward gaining intracellular insight into tumor cell subpopulations and requires unique considerations for optimal platform performance.

## 2. Theory of Dielectrophoresis and Impedance

To study whether high-frequency DEP may be used to detect and sort tumor cell populations with altered DNA content, we first considered the theory underlying DEP. Dielectrophoresis is the polarization of a dielectric particle, in this case a cell, when exposed to a non-uniform electric field. The movement of the particle is induced when the force resulting from polarization is greater than the other forces in the system (e.g., drag force). The DEP force ($F_{DEP}$) is theoretically defined as:(1)$$F_{DEP}=2\pi r^{3}\epsilon_{m}Re\left[CM\left(\omega \right)\right]\nabla {\left|E_{RMS}\right|}^{2}$$
where *r* is the radius of the particle, $\epsilon_{m}$ is the permittivity of the media, *E* is the electric field, and CM is the Clausius–Mossotti factor describing the relationship between the particle and its surrounding media as a function of frequency (ω). The CM factor for a homogeneous sphere is defined as:(2)$$CM\left(\omega \right)=\frac{\epsilon_{p}^{*}-\epsilon_{m}^{*}}{\epsilon_{p}^{*}+2\epsilon_{m}^{*}}$$The pertinent electrical permittivities in a DEP system can be defined as:(3)$$\epsilon_{j}^{*}=\epsilon_{j}-i\frac{\sigma }{\omega }$$
where *j* is used to denote the descriptor: particle, *p*, and media, *m*. The conductivity of the descriptor is σ; ω is the frequency; *i* is the imaginary quantity $\sqrt{-1}$. For more complex systems, such as a nucleated cell, the above representation of the CM factor can be expanded into a double-shell form to model concentric membrane-bounded bodies and is more relevant at frequencies above 1 MHz [33]. The expanded derivation of the CM factor for the double-shell model can be found elsewhere [34].

The CM factor dictates the direction of the DEP force acting on the cell. According to the CM factor, where the particle is more polarizable than the surrounding medium, the particle will experience positive dielectrophoresis (pDEP) and move towards the area of highest electric field gradient. On the contrary, if the surrounding medium is more polarizable than the particle, the particle will experience negative dielectrophoresis (nDEP) and be repelled by the areas of highest electric field gradient. The frequency at which the CM factor of the particle is zero is known as the crossover frequency ($f_{x0}$). For mammalian cells represented using a double-shell model, the CM factor suggests there are two crossover frequencies—that is, where the cell experiences the transition of nDEP to pDEP ($f_{x1}$) and vice versa ($f_{x2}$). In this study, we explored the response of cells as the second crossover frequency is approached.

Under this experimental setup, where the cells are undergoing continuous flow, the DEP force is balanced by the drag force, $F_{Drag}$. The drag force of a spherical particle in a flow stream is defined as:(4)$$F_{Drag}=6r\pi \eta \left(v_{p}-v_{m}\right)$$
where η is the fluid viscosity, $v_{p}$ is the velocity of the particle, and $v_{m}$ is the velocity of the media relative to the laboratory frame of reference. In an ideal system depicted in Figure 1, when the electric field is off, cells with different bio-electrical properties (gray and purple in the figure) flow unhindered with the media. When the electric field is on, however, $F_{DEP}$ is greater than $F_{Drag}$ for one cell type, allowing it to be trapped along the electrodes, while the other cell type remains uninhibited by the DEP force. The coupling of continuous flow and dielectrophoresis allows for continuous characterization and separation (Figure 1b,c).

To determine the electrical behavior of our system, we modeled the impedance change in our device as frequency increases. Impedance, *Z*, can be defined as:(5)$$Z=\frac{V_{s}}{\left|I\right|e^{j\phi }}$$
wherein impedance includes real and imaginary components and is measured based on the current obtained under an applied voltage stimulation, $V_{s}$ is the supplied voltage, *I* is the current, and ϕ is the phase of the signal described by
(6)$$\phi =\arctan(\frac{Re[Z]}{Im[Z]})$$

The theoretical voltage reaching the cells in the system ($V_{out}$) is defined as:(7)$$V_{out}=V_{s}\left|\frac{Z}{Z+50}\right|$$
where the 50 Ω resistance of the cable is accounted for. This model characterizes how the impedance and delivered voltage change with frequency.

## 3. Materials and Methods

### 3.1. Cell Lines and Culture Conditions

DLD-1 cells (ATCC CCL-221), obtained from the American Type Culture Collection (ATCC, Manassas, VA, USA), were maintained in RPMI 1640 with ATCC modification (Thermo Fisher Scientific—Gibco, CA, USA) containing 10% fetal bovine serum (FBS) (Thermo Fisher Scientific) and 1% antibiotic–antimycotic (Thermo Fisher Scientific). Cells were kept in a humidified incubator at 37 ${}^{{}^{\circ}}$C and 5% CO${}_{2}$ and regularly monitored for mycoplasma infection by DNA staining.

### 3.2. Generation and Characterization of Tetraploid DLD-1 Clones

Tetraploid (4N) DLD-1 clones were generated and characterized as described previously [32]. Briefly, diploid (2N) cells were treated with 1.5 μg/mL dihydrocytochalasin B (DCB) (Sigma Aldrich, St. Louis, MO, USA) for 20 h, and limiting dilution in 96-well plates was used to isolate and expand the 4N clones from single cells. Metaphase spreads were collected from 4N clones, and chromosome counting was performed to confirm ploidy/DNA content. G2-synchronized cells (9 μM RO-3306 for 18 h) were fixed with ice-cold methanol for 10 min and 5 μM Cell Tracker Green CMFDA Dye (Thermo Fisher Scientific), and 300 nM DAPI was used to label the cytoplasm and nucleus, respectively, of each cell. Z-stack images spanning the entire height of individual cells were acquired with 0.6 µm steps using a swept field confocal system on a Nikon Eclipse TE2000-U (Nikon Instruments Inc., Melville, NY, USA) inverted microscope equipped with a 60× objective. Cell and nuclear volume analyses were performed in FIJI using a macro for three-dimensional reconstruction [35]. From the diploid cells (2N), two small tetraploid clonal lines (S1 and S2) and two large tetraploid clonal lines (L1 and L2) were generated and used throughout the experiment. The cell and nuclear radii were calculated from volume measurements assuming a spherical shape when in suspension.

### 3.3. Clausius–Mossotti Factor Simulation

A Clausius–Mossotti simulation software, MyDEP, was used to model the impact of radius and nucleoplasm conductivity on the dielectrophoretic behavior [36]. To compare the impact of size, all electrical properties remained the same, and a parameter sweep of the cell radius was performed (7.5, 8.5, 9.5 μm). Another simulation was performed holding all parameters constant and performing a sweep of the nucleoplasm (*np*) conductivity ($\sigma_{np}$ = 0.5, 1.5, 2.5 S/m). The change in nucleoplasm conductivity was used to reflect the expected impact of varying DNA content. While the dielectric properties of the DLD-1 cell line and the isogenic clones used here are not widely characterized in the literature, the simulation parameters were selected from a wide variety of studies summarized in [33]. All simulation parameters for the membrane thicknesses (*th*), cell membrane (*cm*), cytoplasm (*cp*), nuclear envelope (*ne*), and nucleoplasm (*np*) can be found in Table 1.

### 3.4. Sample Preparation

Each experiment was performed by mixing the 2N population and a 4N clone at equal proportions. The two cell populations were labeled with cell-permeant dyes of different colors, namely Calcein Green or Calcein Red-Orange (Invitrogen, Waltham, MA, USA), to aid visualization. Labeling was randomized to negate any effects the cytoplasmic stain might have on each population. Each population was suspended in CytoBuffer (CytoRecovery, Inc., Blacksburg, VA, USA), a low-conductivity buffer solution (σ = 76 μS/cm), at a concentration of 2 × 10${}^{6}$ cells/mL. The samples were mixed (1:1) and loaded into a 1 mL glass syringe (Hamilton Company, Reno, NV, USA).

### 3.5. Device Description and Experimental Setup

The device used in this study was a Metrohm AU10 (Metrohm USA, Riverview, FL, USA) gold interdigitated electrode array with 10 μm spacing between each digit. A straight, 1 mm-wide, and 50 μm-tall polydimethylsiloxane (PDMS) channel with a single inlet and outlet was plasma bonded to the electrode area. The syringe containing the sample was interfaced with the device using Cole-Parmer PTFE Masterflex #30 tubing (Antylia Scientific, Vernon Hills, IL, USA). A Tektronix AFG31000 Series Arbitrary Function Generator was used to energize the electrodes (Tektronix, Beaverton, OR, USA). A schematic of the experimental setup can be seen in Figure 1a. The device was primed with ethanol then buffer through the channel to lower the surface tension in the channel and prevent bubble formation when the sample entered the channel. The syringe containing the sample was mounted on a Harvard Apparatus PicoPump (Holliston, MA, USA) and loaded into the channel at 1 μL/min for 10 min before beginning the experiment. The device was mounted onto a Leica DMi8 inverted microscope (Leica Biosystems, Wetzlar, Germany) for visualization.

### 3.6. Experimental Protocol

The mixed sample was flowed through the channel at 1 μL/min. A sinusoidal signal with a supplied amplitude of 6 Vpp at randomized frequencies between 10 and 100 MHz at increments of 10 MHz energized the electrodes. At each frequency, the field was turned on and off at least three times for multiple data points in each experimental set. These experiments for each diploid and clone pair were repeated at least three times. OBS Studio 29.1.3, a screen-recording software, was used to record the experiment.

### 3.7. Data Analysis

#### 3.7.1. Calculating the Trapping Factor

The video file of the experiment was spliced into segments based on frequency and converted to an image stack using VLC. An ImageJ macro, described elsewhere [37], was used to count the total number of cells and the number of stationary cells of each color (red or green) in each frame. Here, the trapping factor was defined as the average proportion of cells of each color that were stationary when the electric field was turned on. The trapping factor was normalized to any fouling occurring in the device prior to the field being activated, yielding the proportion of cells that were trapped due to the field.

#### 3.7.2. Statistics

An ANOVA with multiple comparisons performed with a Tukey test was used to evaluate the statistical significance throughout this study. Asterisks throughout the study correspond to the following *p*-values: * for *p* < 0.05, ** for *p* < 0.01, *** for *p* < 0.001, and **** for *p* < 0.0001. For cell and nucleus radii comparisons, the sample sizes were as follows: n = 35 (2N), n = 44 (S1), n = 28 (S2), n = 39 (L1), and n = 27 (L2). For trapping factors the sample sizes were as follows: n = 43 (2N), n = 12 (S1), n = 10 (S2), n = 12 (L1), and n = 15 (L2).

### 3.8. Modeling the Impedance Effects of High Frequency on the Electric Field Gradient

A 3D model in COMSOL Multiphysics 6.1 was used to simulate the current, impedance, output voltage, and field gradient ($\nabla E^{2}$) with respect to increasing frequency. The 3D geometry consisted of a glass substrate, gold electrodes, and a low-conductivity buffer sample channel bounded by PDMS. A free tetrahedral mesh with minimum and maximum element sizes of 0.5 μm and 100 μm, respectively, was determined by refining the mesh elements using scaling of the element number until a difference in electric field gradient result was <1% when compared to that of a finer mesh. A 3D cut line was drawn on the top face of the electrode array to extract the field gradient. Along one electrode finger, the maximum value of the field gradient at each frequency was recorded and used to determine the impedance and delivered voltage.

## 4. Results and Discussion

To determine if high-frequency DEP can distinguish tumor cells based on size and ploidy, it was necessary to understand how both changes in cell/nuclear size and DNA content affect the dielectric behavior of cells at high frequencies. To this end, we used our recently developed isogenic cell lines in which the cell and nuclear size doubled in the large 4N clones, consistent with the change in ploidy, while the small 4N clones only increased in volume by 30% compared to the diploid (2N) DLD-1 cells (Figure 2) [32]. Cells from 2N, S1, S2, L1, and L2 clones had calculated radii of 7.46, 8.12, 8.17, 9.14, and 9.49 μm, respectively. The calculated radii of their nuclei were 5.82, 6.48, 6.48, 7.31, and 7.37 μm, respectively. As a result of WGD, the increase in negative charges from the DNA concentrated in the nucleus of the 4N cells is expected to alter the electrical properties of the nucleoplasm compared to the 2N cells, which should be detectable using DEP at high frequencies. Using 2N cells, small 4N clones, and large 4N clones, we examined how cell size (large 4N clones vs. small 4N clones), nuclear content (2N cells vs. small 4N clones), and the coupling of size and nuclear content (2N cells vs. large 4N clones) contribute to cell behavior in a non-uniform electric field.

### 4.1. Clausius–Mossotti Factor Simulations Predict Dielectric Differences between 2N Cells and 4N Clones at High Frequencies

Observing the changes of the CM factor as a result of increasing cell radii or nuclear conductivity provided valuable insight into the anticipated behavior of the diploid cells and tetraploid clones. This analytical approach allowed for the isolation of how each parameter might contribute to the polarization of each cell type with frequency. Given the isogenic system, the perturbation of the CM factor as a result of changing either the cell radius or nuclear conductivity provides a basis for the interpretation of the experimentally observed behaviors.

As evidenced by Equation (Equation 1), the cell radius affects the magnitude of the DEP force experienced by the cell, but it only will directly affect the CM factor of the cell at low frequencies. As seen in Figure 3a, the electrical properties were held constant, and only cell radius was changed. This simulation yielded pronounced differences between the varying cell sizes at low frequencies alone (<100 kHz), indicating that the radius of the cell will affect the polarization of the cell directly before, at, and directly after its first crossover frequency. Figure 3a also suggests that cells with different radii will not experience a difference in polarization due to their size at high frequencies.

On the contrary, when cell radius was held constant and changes in nuclear conductivity were considered, the pronounced differences in CM factors appeared at high frequencies (>10 MHz) (Figure 3b). Further, the second crossover frequency increased as nucleoplasm conductivity increased, meaning that cells with a higher nucleoplasm conductivity will be more polarizable at high frequencies than cells with lower nucleoplasm conductivities. This suggests that distinct electrical behaviors observed at high frequencies experimentally are providing insight into the cell interior, rather than the effect of size alone. While there are no reports establishing a direct link between ploidy and nucleoplasm conductivity, we hypothesized that more DNA, a negatively charged molecule, and the density at which it is packed into the nucleus increase the conductivity of the nucleoplasm. This simulation supports the hypothesis that cells with differing nuclear conductivities as a result of ploidy and nuclear size will be distinguishable at high frequencies. While the cells investigated here also differ in size, cell size effects are dominant at low frequencies.

### 4.2. High-Frequency DEP Reveals Distinct Signatures between 2N Cells and 4N Clones

Here, we used DEP to distinguish cells based on size and nuclear properties. Large 4N clones, small 4N clones, and 2N cells were characterized using high-frequency DEP under flow conditions. Fluid flow not only provided a drag force for enhancing the selectivity of the device, but also acted as a proof-of-concept setup for batch separations of tumor cell subpopulations.

To determine the behavior of cells with dissimilar size and nuclear content, the trapping behaviors of large 4N clones and 2N cells were quantified. The large 4N clones are larger in cell and nuclear size and have twice the amount of DNA content when compared to 2N cells. The dissimilarities in size and nuclear content were both significant contributors to the dielectrophoretic behavior of cells in a non-uniform electric field. In Figure 4a, L2 had a lower trapping factor than the 2N cells at 50–70 MHz. Although L2 clones were twice as large as the 2N cells, they were less susceptible to being trapped at 50–70 MHz. We hypothesize that, similar to what was observed in CM factor simulations (Figure 3b), L2 clones are less polarizable at 50–70 MHz, suggesting a lower nucleoplasm conductivity than 2N cells.

As seen in Figure 4c, S2 clones showed distinct trapping behavior from 2N cells at 70 MHz. The significant change in trapping factor illustrated the contribution of intracellular electrical properties to the dielectrophoretic behavior since size was not a significant factor between these two populations. S2 had a higher trapping factor at this critical frequency, suggesting higher polarizability than 2N cells. Higher polarizability at high frequencies can be attributed to higher nuclear conductivity (Figure 3b). Although similar in size, S2 exhibited distinct polarizability when compared to the 2N cells as a result of increased nuclear conductivity from tetraploidy. This selective separation of cells similar in size but different in nuclear composition reveals that high-frequency DEP can be used to investigate and distinguish tumor cell subpopulations based on nuclear content.

A general trend observed from this experiment was that small 4N clones (S1 and S2) tended to have a higher trapping factor than 2N cells, while large 4N clones had a lower trapping factor than 2N cells. Due to the contribution of the size to the DEP force and the drag force, a quick assumption would be that the drag force is much greater on large 4N clones due to their size, and thus, they are less susceptible than small 4N clones or 2N cells to succumb to the DEP force. The small 4N clones, however, while not statistically larger, were on average 1.3× the size of their 2N ancestor and displayed a higher trapping factor than both 2N cells and large 4N clones (Figure 4b). If drag force were the main contributor to the decreased trapping factor of large 4N clones, small 4N clones would experience a similar decreased trapping factor with respect to the 2N cells as a result of increased drag acting on the radius of the cell. Despite their larger size, small 4N clones had a higher trapping factor than 2N cells. This suggests that the differences in the trapping factor are representative of their intrinsic electrical properties. We hypothesize that large 4N clones would experience no DEP force at a $f_{x2}$ that is lower than that of 2N cells or small 4N clones. While not directly comparable, the CM factor of large 4N clones suggests that these cells will have a lower $f_{x2}$ than the 2N cells or small 4N clones due to a decrease in nucleoplasm conductivity.

### 4.3. Modeling the Impedance Effects of High Frequency on the Electric Field Gradient

As seen in Figure 4, the trapping factor decreased significantly as the frequency increased past 40 MHz. While approaching the second crossover frequency, theoretically anticipated to be closer to 200 MHz, the DEP force was not expected to decrease drastically at this frequency range. Because of this apparent phenomenon in the data, the impedance of the device at these frequencies was simulated to characterize the device performance.

From the simulation, the field gradient at the edge of each electrode across the array changed two orders of magnitude within the relevant frequency range (10–100 MHz) (Figure 5a). As the frequency increased, the impedance and the voltage reaching the cells in the device drastically decreased. The phase of the signal also changed as the frequency increased, leading to a further degradation of the signal. As a result, the field gradient across the electrode array decreased with increasing frequency. The maximum field gradient across the electrode array for each frequency is plotted in Figure 5b. With a decrease in field magnitude, the overall magnitude of the DEP force also decreased with frequency, evidenced by the decrease in the trapping factor after 40 MHz. This decay of the signal continued to worsen as the frequency approached 100 MHz when compared to the 1 MHz signal. The amplitude of the signal dropped below approximately 1 V beyond 40 MHz, greatly reducing the trapping capabilities of the system (Figure 5c). Signal degradation characterized by impedance analysis as a result of parasitic capacitance at high frequencies revealed a field gradient magnitude threshold below which trapping was less likely; for this particular platform, that threshold lied below 6.8 × 10${}^{12}$ V${}^{2}$/m${}^{3}$. Here, parasitic capacitance depends on the conductivity of the medium and ion mobility.

The simulation of the impedance of this platform, in tandem with the experimental data, illustrated challenges associated with high-frequency DEP. Small spacing between electrodes might increase the field gradient magnitude without the need for complex electronic systems. This solution, however, might also increase the likelihood of capacitive losses, as seen here. It is imperative to account for impedance at high frequencies. Signal degradation will significantly affect the performance of the device and, as a result, the cell behavior in response to DEP forces. In this case, the decrease in the field gradient due to a change in impedance created a more-efficacious force balance between the DEP force and drag force. While the DEP force was decreasing with frequency, the drag force was able to compete more significantly. This force balance revealed the distinct signatures observed in Figure 4.

## 5. Conclusions

Technologies that identify and enrich aneuploid cells and/or their 4N precursors from a mixed population are necessary to understand their contribution to age-related diseases, such as cancer. In this study, high-frequency DEP was used to characterize human CRC cells that differ in size and/or DNA content directly affecting the nuclear charge dilution. While differing dielectrophoretic behaviors were observed under appropriate coupling of dielectrophoretic and drag forces, the field intensity greatly decreases as the frequency approaches 100 MHz. While high-frequency DEP is a promising regime according to theory, controlling for impedance losses to maintain a consistent field gradient is difficult and compounds the dielectrophoretic results. This study demonstrated the feasibility of high-frequency DEP as a tool to identify and distinguish cells with varying ploidy in a heterogeneous sample, but high-frequency phenomenon, such as changes to the impedance of the system, should be managed.

## Figures and Tables

**Figure 1 micromachines-14-01723-f001:**
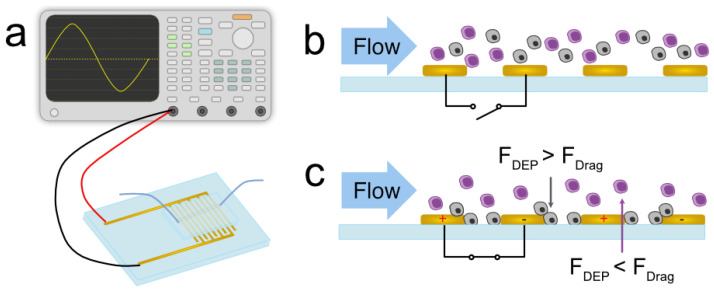
(**a**) Experimental setup with the generator to the gold interdigitated electrode array. (**b**) Cells travel unhindered when the field is turned off. (**c**) When the electrodes are energized, cells with distinct bio-electric signatures will exhibit differing behavior where the DEP force overcomes the drag force at a higher proportion for one cell type, but not the other.

**Figure 2 micromachines-14-01723-f002:**
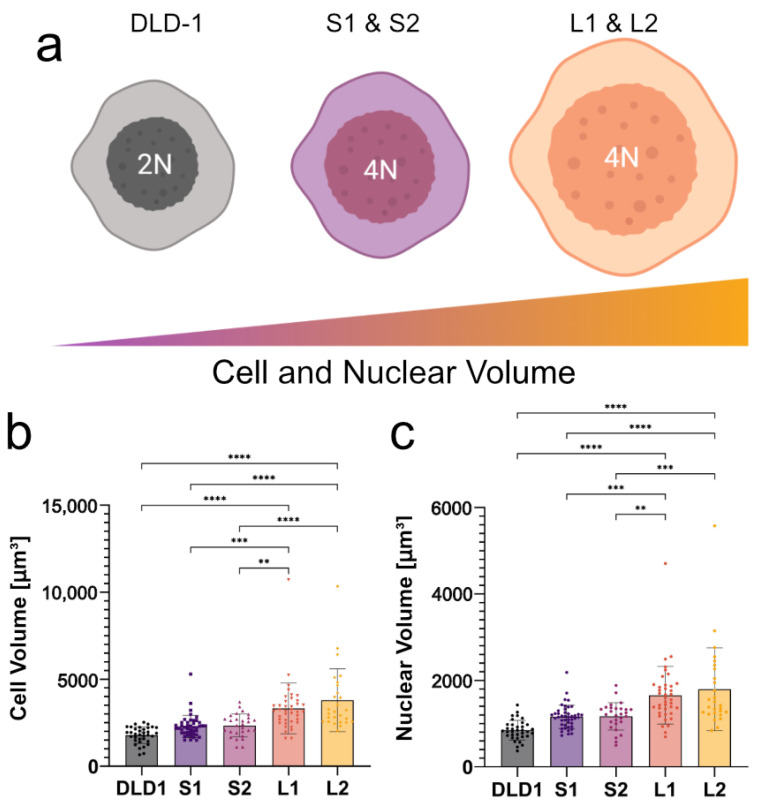
2N cells and 4N clones vary in cell and nuclear size. (**a**) Cell and nuclear size increases sequentially from 2N to the small 4N clones to the large 4N clones. (**b**) Cell volumes of the small 4N clones are approximately 1.3× that of the diploid. The large 4N clones are approximately 2× the size of the diploid, scaling with ploidy, and are significantly larger than 2N cells and small 4N clones. (**c**) The nuclear volumes of 2N cells, small 4N clones, and large 4N clones show a similar trend. Significant p-values are denoted as ** for *p* < 0.01, *** for *p* < 0.001, **** for *p* < 0.0001.

**Figure 3 micromachines-14-01723-f003:**
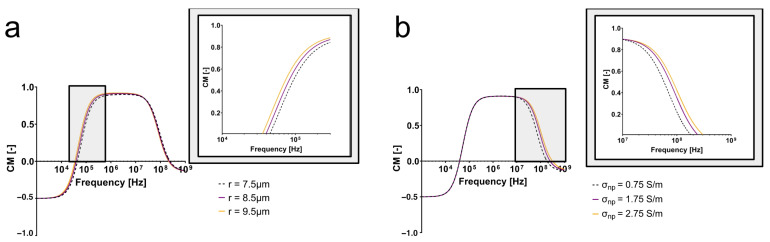
Clausius–Mossotti factor simulations suggest that high-frequency DEP can exploit intracellular differences. (**a**) Size affects the electrical properties of the cell adjacent to its first crossover frequency. (**b**) Changes in nuclear properties are distinguishable at high frequencies.

**Figure 4 micromachines-14-01723-f004:**
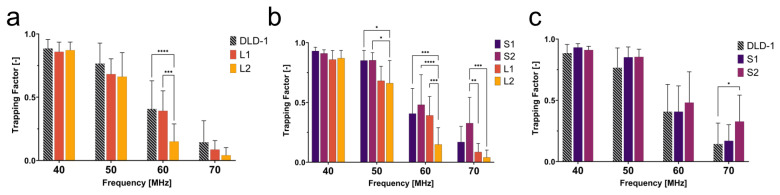
2N cells and 4N clones have distinct trapping behavior at high frequencies. (**a**) Large 4N clones and 2N cells differ in size and nuclear content and exhibit differing bio-electric signatures at 50–70 MHz. (**b**) Small and large 4N clones differ only in size and charge density and exhibit differing bio-electric signatures at 50–70 MHz. (**c**) The 2N cells and small 4N clones are similar in size, but different in charge density. This results in a much finer regime of bio-electrical differences at 70 MHz. Significant p-values are denoted as * for *p* < 0.05, ** for *p* < 0.01, *** for *p* < 0.001, **** for *p* < 0.0001.

**Figure 5 micromachines-14-01723-f005:**
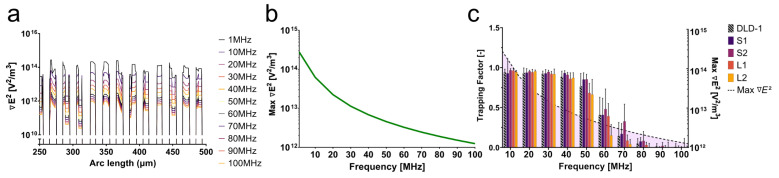
In response to changes in impedance, the field gradient also changes as a result of decreased voltage. (**a**) The modeled field gradient across the electrode array decreases as a function of frequency. (**b**) The maximum field gradient across a single electrode finger decreases with increasing frequency. (**c**) The decrease in the trapping factor corresponds to the decrease in the field gradient.

**Table 1 micromachines-14-01723-t001:** Parameters to model the Clausius–Mossotti factor of diploid and tetraploid clones.

Variable	Radius Simulation	Nuclear Conductivity Simulation
*r* (μm)	7.5, 8.5, 9.5	7.1
$r_{n}$ (μm)	5.64	5.64
$\sigma_{cp}$ (S/m)	0.8	0.8
$\epsilon_{cp}$	50	50
$th_{cm}$ (nm)	7	7
$\sigma_{cm}$ (S/m)	1.4 × 10^−9^	1.4 × 10^−9^
$\epsilon_{cm}$	6.32	6.32
$\sigma_{np}$ (S/m)	1.35	0.5, 1.5, 2.5
$\epsilon_{np}$	52	52
$th_{ne}$ (nm)	40	40
$\sigma_{ne}$ (S/m)	1.1 × 10^−3^	1.1 × 10^−3^
$\epsilon_{ne}$	28	28

## Data Availability

The data presented in this study are contained within the article.

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
