# Peer review of "High-Frequency Dielectrophoresis Reveals That Distinct Bio-Electric Signatures of Colorectal Cancer Cells Depend on Ploidy and Nuclear Volume"

_micromachines, 2023, doi:10.3390/mi14091723_

Round 1

Reviewer 1 Report

Title: High-frequency dielectrophoresis reveals that distinct bioelectric signatures of colorectal cancer cells depend on ploidy and nuclear volume.

Overall results of review: Modest revision needed. There are several key pieces of information missing that need to be addressed or included before suitable for publication. The authors are commended for their efforts to merge theory, simulation, and difficult experimentation. Authors are encouraged to pay particular attention to feedback pieces #: 6, 7, 12, 15, and 16 in a revision.

1.          P2, L40-46: This paragraph slightly inaccurately describes the “traditional” DEP regimes and the association of alpha, beta, and gamma dispersions. The statement that DEP can separate cells based on differences in dielectric dispersion is accurate, but typically DEP leverages interfacial polarizations up to MHz frequencies. Alpha and beta dispersions capture these effects when permittivity is determined based on a whole cell, and the effects of varying permittivity and conductivity of subcellular structures are ‘smeared’ out over the whole cell. Gamma dispersions are traditionally in the GHz regime and associated with polarization of water molecules.

2.          P2-3: The definition of DEP force, Clausius-Mossotti factor, etc in Eqn 1, 2, and 3 here is correct for a homogeneous sphere. This should be so indicated in the text. The suggestion (P3, L93) that there are two cross over frequencies is only true if there are 2 or more interfaces between homogenous regions of different complex permittivities (as is the case in a shell model). This should also be indicated, with citation.

3.          Eqn 4: velocities are defined relative to the lab frame of reference, this should be so indicated.

4.          Eqn 6: While the phase of the complex impedance, relative to the phase of the applied voltage, is accurate, it’s not really properly derived from the phase of the impedance. A quantity can’t be defined in terms of itself. More accurately, this should be the phase of the measured current. In this definition, is the current a complex quantity? Or is the reason for the absolute value sign to correct for negative impedance?

5.          P3-4: It would be helpful to identify Z as a complex valued impedance somewhere. V_s is technically also complex, but can be assumed real without significant loss of generality. Unless it is spatially non-uniform on a lengthscale approaching the wavelength of applied frequencies?

6.          SS 3.3 and Table 1: Where did these numbers come from? Were they measured? Taken from another source? Assumed for the sake of simulation? If the last, there should be some justification provided as to the accuracy of chosen values. Cross-over frequencies are non-unique in the multishell model (ie, different sets of values for shell electrical properties can produce the same cross-over frequency).

7.          SS 3.3 and Table 1: are the values of nuclear conductivity cited here consistent with other measurements? Can the change in nuclear conductivity with increasing ploidy be supported with data or literature?

8.          SS 3.5: How wide was the microchannel?

9.          SS 3.6: were frequencies randomized at all to account for carryover between responses at different frequencies. (i.e., trapping at a 10MHz leaves cell membrane residue on the electrode array, making it easier for cells to trap at 20MHz?) Randomizing the order of applied frequencies would ensure that the observed phenomena are a result of the frequency under test.

10.    P6, L209: What do the designations S1, S2, L1 and L2 refer to?

11.    P7, L228: Authors state that changes in cell size only affect fCM “in close proximity to its first cross-over frequency.” It is unclear what is meant by “close proximity” and “first” terms. This reviewer assumes that “close” means frequencies that are around a factor of 2 spanning the cross over frequency, and that “first” means lowest frequency, typically corresponding to the interfacial polarization between the cell membrane and the surrounding media.

12.    Figure 2: the results here show a significant change in nuclear volume with increasing amounts of nuclear material. Could this also be simulated in MyDEP?

13.    P8, L244: Authors state “Cells of dissimilar size … will be distinguishable at high-frequencies…” but figure 3 and the main hypothesis seems to indicate that this should be “at LOW frequencies”.

14.    P9, L266-7: Suggest combining these sentences. “Albeit” in this context would be better replaced by “Although” and “portrayed” would be better replaced by “exhibited”.

15.    Figure 5: The details of the COMSOL model used to extract these data are too sparse to evaluate. Due to the highly nonlinear nature of the DEP force and it’s dependence on the gradient of a squared field, the results will be highly dependent on mesh resolution. Without information on such a study, it is difficult to trust the resulting gradient values. If the only information sought from the study is a general trend, the such should be stated. However, the results – the curve in figure 5B – does not align with the data in figure 5C, suggesting that whatever the causal effect of the observations in figure 5b, it does not explain the observations in 5C. Even though an effect is observed, no attempt is made to explain why the gradient would decrease with increasing frequency.

16.    Figure 5C: double layer capacitance may contribute to effects in this frequency range, depending on the conductivity of the medium and the size of ions contributing to co- and counter-ion mobilities.

Reviewer 2 Report

The authors present an excellent experimental/theoretical study of cells of mixed ploidy, with the aim of developing a separation strategy.  The paper is highly novel, scientifically rigorous and well-written.  As such, the paper is definitely worthy of publication in Micromachines.  I would suggest the following small changes that the authors might consider.

1. The number of repeats needs to be explicitly stated if the statistics are to make sense.  I suggest n numbers be included in both the Materials section and in the figure legends where significance is indicated.

2. Given the significance of separation to the rationale of the work, the paper would benefit from some discussion of this.  For example, Gupta et al.'s paper about the Apostream (Biomicrofluidics 2013) would be a useful starting point for the separation of cancer cells, and would allow a reference to practical separation mechanisms. 

3. It would also be useful to discuss the practicalities of separation at high frequencies.  One of the reasons there has been little work on this is that it necessitates signal generators delivering quite substantial power (in order to drive large numbers of electrodes) at high frequencies, which can be difficult to achieve in the face of capacitive loss, and difficult to engineer given the required slew rates.  Clearly, solving these problems are beyond the remit of this paper, but some discussion of the difficulties in cell separation in the tens of MHz would be appropriate.  
